# Do parents counter-balance the carbon emissions of their children?

**Jonas Nordström**[1]*, **Jason F. Shogren**[2], **Linda Thunström**[2]*

**1** Sweden and Department of Food and Resource Economics, Lund University School of Economics and Management, University of Copenhagen, Copenhagen, Denmark, **2** Department of Economics, University of Wyoming, Laramie, WY, United States of America

\* jonas.nordstrom@agrifood.lu.se (JN); lthunstr@uwyo.edu (LT)

**Data Availability Statement:** The household expenditure data (from the Household budget survey 2008 and 2009) that is used for this article is provided by Statistics Sweden. In Sweden, data for individual respondents (microdata) are

## Abstract

It is well understood that adding to the population increases $CO_2$ emissions. At the same time, having children is a transformative experience, such that it might profoundly change adult (i.e., parents') preferences and consumption. How it might change is, however, unknown. Depending on if becoming a parent makes a person "greener" or "browner," parents may either balance or exacerbate the added $CO_2$ emissions from their children. Parents might think more about the future, compared to childless adults, including risks posed to their children from environmental events like climate change. But parenthood also adds needs and more intensive competition on your scarce time. Carbon-intensive goods can add convenience and help save time, e.g., driving may facilitate being in more places in one day, compared to public transportation or biking. Pre-prepared food that contain red meat may save time and satisfy more household preferences, relative to vegetarian food. We provide the first rigorous test of whether parents are greener or browner than other adults. We create a unique dataset by combining detailed micro data on household expenditures of all expenditure groups particularly important for $CO_2$ emissions (transportation, food, and heating/electricity) with $CO_2$ emissions, and compare emissions from Swedish adults with and without children. We find that parents emit more $CO_2$ than childless adults. Only a small fraction of adults permanently choose not to have children, which means any meaningful self-selection into parenthood based on green preferences is unlikely. Our findings suggest that having children might increase $CO_2$ emissions *both* by adding to the population and by increasing $CO_2$ emissions from those choosing to have children.

## 1. Introduction

Population growth increases $CO_2$ emissions, and the decision to reproduce has been found to have the largest impact on the climate, amongst any types of consumer behavior [1, 2]. Naturally, adding more people to the planet increases $CO_2$ emissions at the extensive margin. The open question we explore herein is whether increased $CO_2$ emissions will also occur at the intensive margin–the parents themselves. Will parents change their consumption as a result of

protected by the Secrecy Act. Researchers can
apply for access to microdata for use in specified
research projects. The system for researchers'
access to microdata stored at Statistics Sweden is
called Microdata Online Access (MONA). Users
who receive deliveries via MONA can process the
data over an internet connection without microdata
ever leaving Statistics Sweden. Contact information
to apply for access to (MONA) microdata: e-mail
mikrodata@scb.se, phone +46 10 479 50 00,
website: www.scb.se/en/services/guidance-for-
researchers-and-universities/ The authors had no
special access to data that other researchers would
not be able to obtain.

**Funding:** The author(s) received no specific
funding for this work.

**Competing interests:** The authors have declared
that no competing interests exist.

their reproduction choice? If so, will they attenuate or accentuate the increased emissions of
their children?

Parenthood has the potential to significantly affect climate change. While other consumer
decisions might also affect $CO_2$ emissions, it might be of particular interest to study the effect
on the climate from parenthood. First, as opposed to most other consumer decisions, having a
child changes people–it changes people's "core preferences" [3], therefore potentially also peo-
ple's green preferences. Second, most people become parents, such that any altered green
behavior due to parenthood has the potential to significantly affect total $CO_2$ emissions. Third,
it seems entirely feasible to design policies that could mitigate any increased $CO_2$ emissions
from parenthood, by targeting the unique situation of parents.

Whether parents mitigate or accentuate the increased emissions of their children is *a priori*
ambiguous. Two countervailing forces are at work. First, the transformational experience of
having a child. On having children, Paul (2015), pp.156, clarifies: "a personally transformative
experience radically changes what it is like to be you, perhaps by replacing your core prefer-
ences with very different ones" [3]. Particularly relevant to our context, parenthood makes
people worry more about the future [4,5], and increases the strength of preferences for envi-
ronmental goods [6], which may suggest using less carbon-intensive goods and services to
reduce future risks of climate change [3, 7–12]. But the flip side is that time constraints now
become more binding and convenience may become more important–parents may need to be
in more places in one day. They also need to feed more people. Eating more pre-prepared, red
meat carbon-intensive meals may add convenience and save time. Parents' environmental
behavior may also be motivated by other factors, such as health and income [6,13–15].

Studies provide mixed evidence on whether parenthood changes environmental prefer-
ences and attitudes [16–19]. Previous research based on the Understanding Society Survey
finds some support that having a child reduces a selection of self-reported environmental
behaviors [20]. Similarly, a study based on the World Value Survey finds that parents are less
willing to spend money on improving the environment than non-parents [21]. Although pro-
viding valuable insights, previous studies have only examined a sub-set of behaviors that relate
to environmental concern, and have not addressed the net environmental impact from
parenthood.

In this paper, we empirically examine if parents leave a smaller carbon footprint, compared
to non-parent adults. To perform our analysis, we combine detailed data on household expen-
ditures with $CO_2$ emissions, generating a uniquely detailed micro level dataset that spans all
major household expenditure groups associated with $CO_2$ emissions. Our data allows us to
quantify and compare $CO_2$ emissions from Swedish adults with and without children.

Our analysis relies on cross-sectional observational data. While the benefits of our data are
the level of detail and the wide range of consumption types included in the data set, the draw-
back is that we cannot control for any potential self-selection into parenthood that is based on
green preferences, which undermines our ability to suggest any causal effect between parent-
hood and green behavior. In theory, such self-selection could exist, i.e., people who are less
"green" might self-select into parenthood, which in turn might bias our results. However,
empirical evidence suggests self-selection into parenthood in general (i.e., not only as a result
of any potential environmental concerns) is uncommon, and self-selection based on reasons
that affect $CO_2$ emissions is even more unlikely.

Some of the childless households in our data will likely have children later in life, some are
involuntarily childless (just like some households with children likely did not plan to have chil-
dren), while some actively chose to remain childless. Any self-selection affecting our results
only pertains to the latter group–those choosing never to have children. This is, however, a
small group–of childless households age 20–40, only five percent state they do not want to

have children [22]. The average young Swedish childless two adult-household plans to have children in the future. Further, over 50 percent of older childless households said they tried, but failed to have children. Of particular relevance to our analysis, not all willingly childless households (i.e., all of the five percent of 20–40 year old childless households) choose to abstain from having children due to reasons that might affect $CO_2$ emissions, which is the kind of selection we would worry about for our study. In fact, the main reasons against having children have been found to be emotional stress from having a child, career interference, and not wanting the responsibility of bringing up a child, while environmental concern (e.g., concern for overpopulation), or variables that might correlate with $CO_2$ emissions (e.g., travel preferences), have not been found to significantly affect the decision to be childless [23]. Still, a cautionary note is granted. Although unlikely, self-selection into parenthood in our data that is based on, or correlates with, green preferences cannot be ruled out, meaning that we can say with confidence that our analysis succeeds in comparing $CO_2$ emissions between parents and non-parents, while there is some uncertainty surrounding its success at identifying the effect of $CO_2$ emissions on *becoming* a parent.

This paper provides the first rigorous test of whether parents themselves are "greener", e.g., have a smaller carbon footprint. We find that being a parent means a Swedish person leaves a larger carbon footprint, mainly driven by higher $CO_2$ emissions from transportation and food consumption. A potential explanation for our results may be that Swedish parents use carbon-based consumption as a substitute for their tighter time constraints. A comprehensive report on time usage by Swedish households, shows that out of all people, parents with small children have the least leisure time, defined as time not spent working, paid or in the household [24]. Mothers and fathers of children 7 years and younger have around four hours of daily leisure time, while daily leisure time of a childless woman age 20–44 is around five hours, and that of a childless man of the same age is about six hours.

Altruistic parents' consumption may also be impacted by the child's immediate preferences for carbon-intense consumption, such as a taste for red meat, flights to family friendly resorts, and so on [25–27]. Children may of course also be concerned about the environment, which in turn may affect household consumption. Even though children in the Nordic countries have been found to be more environmentally concerned than their parents [28,29], there is a substantial gap between their attitudes and actions [30,31].

Our results suggest that parents emit more $CO_2$ than non-parents, and that the greenest adult Swede is childless and lives alone. Our results may be particularly striking, given Sweden is our focus. If the transformation from parenthood causes parents to focus more on the environment, we might expect this behavior to be particularly pronounced in Sweden relative to other developed countries. Most Swedes believe climate change is real and have accepted sizable $CO_2$ taxes–both suggest that reducing one's carbon footprint for their off-springs matters. In addition, households with children are subsidized, which helps to alleviate some of the time crunch for parents. Sweden has generous parental leave, subsidized daycare, and parents have a legal right to reduced work hours [32,33]. But Sweden also has one of the world's highest female labor participation rates (69.5 percent in 2015, as compared 51.4 percent in the European Union and 56.7 percent in the US), which may add to the time constraint of household with children [34].

## 2. Data and econometric specification

We summarize our unique data below, and S1 Appendix provides more specific details, including how to access the data. Our data contained household expenditures on goods and services from a set of 4,000 households surveyed by Statistics Sweden for 2008–2009. However,

the sample used in our analysis is smaller ($N = 2,692$), given we limit our data in a few key ways–we focus on households with a positive disposable income, that have a maximum of two adults (if a household has three or more adults, we lack the necessary information to determine if all adults are parents, or some are adult children). Further, we exclude households in which at least one household member is retired. The last limitation is imposed given we want to minimize the amount of households in our sample consisting of adults who once lived with children but not now, since our analysis builds on the assumption that all households in our analysis that are currently childless also never had children. We explore the sensitivity of our results to even stricter age limits for the households included in our analysis (see S2 Appendix), and find results are robust.

Household characteristics include number of adults, number of children, age of household members, disposable income level, type of housing (apartment, house, ranch), and size of housing (in square meters). Tables 1 and 2 show descriptive statistics on household characteristics, for all households in our sample and sub-groups of households.

Next, we calculate quantities consumed by dividing expenditures by 2009 prices for all expenditure items that constitute the vast majority of household $CO_2$ emissions. (We use prices for 2009 for the sample since we have reliable price data for 2009 and the price changes were minor in Sweden between 2008 and 2009 –the change in the consumer price index (CPI) was -0.3 percent). Third, we match the detailed consumption data with the corresponding $CO_2$ emissions per unit. Classified according to the international COICOP classification system on a four-digit level, we capture detailed data on four expenditure groups: (a) food and

**Table 1. Descriptive statistics.**

| | All households | | Two adults with children | | Two adults without children | |
|---|---|---|---|---|---|---|
| Variable | Mean | Std.Dev. | Mean | Std.Dev. | Mean | Std.Dev. |
| No of children | 1.20 | 1.16 | 2.01 | 0.85 | | |
| No of children age 0–6 | 0.42 | 0.71 | 0.74 | 0.82 | | |
| No of children age 7–12 | 0.36 | 0.65 | 0.60 | 0.76 | | |
| No of children age 13–17 | 0.31 | 0.62 | 0.49 | 0.72 | | |
| No of children age 18–19 | 0.11 | 0.33 | 0.17 | 0.40 | | |
| Two adults w. children | 0.22 | 0.41 | | | | |
| One adult w/o children | 0.17 | 0.38 | | | | |
| One adult w. one or more children | 0.08 | 0.27 | | | | |
| Age oldest in household | 43.74 | 11.21 | 42.20 | 7.67 | 48.43 | 14.08 |
| Disposable income | 429.82 | 231.25 | 513.87 | 220.73 | 452.90 | 209.41 |
| $CO_2$ consumption[a] | 6891.78 | 3,390.53 | 8,230.85 | 3,225.58 | 6,738.16 | 2,928.68 |
| $CO_2$ food | 2,287.74 | 1,398.22 | 2,853.27 | 1,404.37 | 2,109.04 | 1,050.36 |
| $CO_2$ meat | 898.48 | 835.83 | 1,119.54 | 914.09 | 871.17 | 702.72 |
| $CO_2$ transportation | 3,312.00 | 2,591.94 | 4,035.11 | 2,622.32 | 3,394.34 | 2,483.45 |
| $CO_2$ gasoline | 3,168.84 | 2,608.75 | 3,893.71 | 2,640.37 | 3,179.53 | 2,502.84 |
| $CO_2$ electricity & heating | 905.40 | 785.46 | 916.48 | 902.31 | 891.55 | 789.93 |
| Living area (in $m^2$) | 114.68 | 51.22 | 133.37 | 45.32 | 112.09 | 52.69 |
| Cost all-inclusive trip[b] | 14,526.05 | 21,682.23 | 15,875.86 | 23,233.67 | 18,592.28 | 23,158.15 |
| Cost domestic trip[b] | 508.67 | 2,362.67 | 579.79 | 2,727.21 | 574.66 | 2,401.66 |
| Cost international trip[b] | 13,974.72 | 21,235.52 | 15,217.50 | 22,631.24 | 18,017.62 | 22,893.25 |
| $N$ | 2,692 | | 1,422 | | 582 | |

[a] Total $CO_2$ emissions from consumption included in our analysis (food, transportation, electricity and heating, clothing and shoes).
[b] In SEK.

**Table 2. Descriptive statistics, continued.**

| Variable | One adult with children | | One adult without children | |
| --- | --- | --- | --- | --- |
| | Mean | Std.Dev. | Mean | Std.Dev. |
| No of children | 1.71 | 0.72 | | |
| No of children age 0–6[a] | 0.31 | 0.59 | | |
| No of children age 7–12[b] | 0.49 | 0.70 | | |
| No of children age 13–17[c] | 0.65 | 0.74 | | |
| No of children age 18–19 | 0.26 | 0.46 | | |
| Two adults w. children | | | | |
| One adult w/o children | | | | |
| One adult w. one or more children | | | | |
| Age oldest in household | 42.75 | 7.95 | 43.00 | 15.26 |
| Disposable income | 266.54 | 153.73 | 221.59 | 126.88 |
| $CO_2$ consumption[d] | 5,122.84 | 2,301.05 | 3,800. 45 | 2,219.30 |
| $CO_2$ food | 1,791.57 | 1,217.89 | 1,021.86 | 680.99 |
| $CO_2$ meat | 614.66 | 664.28 | 392.59 | 466.56 |
| $CO_2$ transportation | 1969.85 | 1,719.07 | 1,703.40 | 1,922.03 |
| $CO_2$ gasoline | 1,827.92 | 1,746.64 | 1,577.12 | 1,946.73 |
| $CO_2$ electricity & heating | 1,080.31 | 549.68 | 806.98 | 365.16 |
| Living area (in $m^2$) | 97.13 | 36.52 | 69.23 | 38.74 |
| Cost all-inclusive trip[e] | 8,860.53 | 16,295.40 | 7,987.81 | 13,760.61 |
| Cost domestic trip[e] | 267.69 | 1,064.52 | 322.56 | 1,313.79 |
| Cost international trip[e] | 8,578.58 | 16,239.89 | 7,665.25 | 13,580.67 |
| N | 219 | | 467 | |

[a] Eleven percent of the households have 2 or more children.

[b] Eight percent of the households have 2 or more children

[c] Seven percent of the households have 2 or more children. The remaining households have zero or one child in the respective age groups. If we do not divide the children into different age groups, our data show that 42 percent of the households have 2 or more children aged 0–19 years.

[d] Total $CO_2$ emissions from consumption included in our analysis (food, transportation, electricity & heating, clothing and shoes).

[e] In SEK.

non-alcoholic beverages, (b) transportation, (c) clothing, including shoes, and (d) housing. Tables 3 and 4 summarize the expenditure groups and associated levels of $CO_2$ emissions (S1 Appendix provides the specific matching numbers and sources). These four expenditure groups account for about 85 percent of the households' total $CO_2$ emissions. For some expenditure groups, we need to use various sources of price and $CO_2$ emissions data, which might create uncertainty in some of our consumption and emissions data. We therefore perform numerous robustness checks where we vary price and emission assumptions (see S2 Appendix).

Now consider our econometric specification. We examine the effect on adult $CO_2$ emissions from adding children to a two-adult household. Using Ordinary Least Squares (OLS) regression, with heteroscedasticity-consistent (Huber-White) standard errors, we estimate the following model:

*Total $CO_2$ emissions per household (kg) = constant + b1\* Number of children 0–6 years*

*+ b2\*Number of children 7–12 years*

*+ b3\* Number of children 13–17 years*

*+ b4\* Number of children 18–19 years*

*+ b5\* Two adults without children*

*+ b6\* One adult without children*

*+ b7\* One adult with one or more children*

*+ b8\* Age of the oldest person in the household*

*+ b9\* Disposable income*

*+ b10\* Disposable income squared + error term*

**Table 3. $CO_2$ emissions from food.**

| COICOP | Good | Kg $CO_2$ per kg/liter good | COICOP | Good | Kg $CO_2$ per kg/liter good |
|---|---|---|---|---|---|
| | *Bread and cereals* | | | *Oils and fats* | |
| 0111101 | Rice | 2 | 0115101 | Butter | 8 |
| 01112 | Bread | 0.8 | 0115102 | "Diet" butter | 4.8 |
| 0111301 | Pasta | 0.8 | 0115201, | Margarine | 1.5 |
| 0111409 | Sandwich | 1.5 | 0115202, | | |
| 0111501 | Flour | 0.6 | 0115203, | | |
| 0111503 | Cereal | 0.8 | 0115204 | | |
| 0111504 | Cookies | 1 | 01153 | Olive oil | 1.5 |
| 0111505 | Pastries | 1 | 01154 | Cooking oil | 1.5 |
| 0111508 | Pizza | 2 | | Mayonnaise | |
| | *Meat* | | | *Fruit & vegetables* | |
| 01121, 0112501 | Beef | 26 | 0116 | Fruit | 0.52[a] |
| 01122, 0112502 | Pork, bacon etc. | 6 | 0117 | Vegetables | 1[b] |
| 01123 | Sheep | 21 | 01177 | Potato | 0.1 |
| 01124, 0112503 | Poultry | 3 | 0117803 | Potato chips | 2 |
| 0112505 | Brawn | 7 | | | |
| 0112506 | Sausages | 6 | | *Sugar, jam, candy* | |
| 0112507 | Pâté | 7 | 01182 | Jams, marmalades | 3 |
| 0112508 | Charcuterie | 7 | 01183, 01184 | Chocolate, candy etc. | 2 |
| 0112601 | Ready meals | 10.6 | 01185 | Ice cream | 2 |
| | | | 0119401 | Snacks | 1 |
| 0113 | *Fish* | 3 | | | |
| | *Milk, cheese, eggs* | | | *Non-alcoholic beverages* | |
| 01141, 01142 | Milk | 1 | 01211 | Coffee | 3 |
| 0114401 | Yoghurt | 1 | 01221 | Table water | 0.3 |
| 0114402 | Sour milk | 1 | 01222 | Soda | 0.3 |
| 011460101, 011460102, 011460103 | Cream | 4 | 01223 | Fruit juices | 3 |
| 011450101, 011450102 | Cheese | 8 | | | |
| 0114701 | Eggs | 2 | 111 | Restaurants | [c] |
| 0114502, 0114503, 0114504, 0114603 | Other dairy products | 2 | | | |

[a]$CO_2$ emissions for fruit produced in the Nordic countries = 0.2 kg/(kg fruit), for imported fruit from non-Nordic countries = 0.6. kg/(kg fruit). 20% of the fruit consumption is assumed to be domestically produced, corresponding to the expenditure share on apples.

[b]$CO_2$ emissions for domestically produced root crops = 0.2 kg/(kg root crop); vegetables produced in the Nordic countries = 1.0 kg/(kg vegetables); imported vegetables from non-Nordic countries = 1.4 kg/(kg vegetables); imported vegetables with aircraft = 11 kg/(kg vegetables).

[c]$CO_2$ emissions based on expenditure shares [35].

**Table 4. $CO_2$ emissions from clothes, electricity, heating and transportation.**

| COICOP | Good | $CO_2$ | COICOP | Good | $CO_2$ |
|--------|------|--------|--------|------|--------|
| 03 | Clothing and footwear | d | | Transport | |
| | Electricity and heating | | 0722101, | Petrol/Diesel | 2.24 kg/liter |
| 0451 | Electricity | 20 kg/MWh | 0722102 | | (2.6 kg/liter) |
| 0453 | Liquide fuels (oil) | 2.69 kg/liter | 0731 | Railway | 1.1 g/passenger km |
| 0454 | Solid fuels (pellets) | 6 kg/MWh | 0732 | Bus, Taxi | 79 g/passenger km |
| 0455 | District heating | 92.7 kg/MWh | 0733 | Air | 130 kg per trip [e] |
| | | | 0734 | Boat | 2.24 kg/liter gasoline [f] |
| | | | 0735 | Combined transport | 30 g/passenger km |

[d] $CO_2$ emissions based on expenditure shares [31].

[e] Households with positive expenditures on air travel are assumed to emit 130 kg of $CO_2$, corresponding to a round trip Stockholm-Gothenburg (the two largest cities in Sweden).

[f] Expenditures are assumed to be on gasoline. In the sensitivity analysis we assume all households with positive expenditures on boat travel have made a round trip Stockholm-Helsinki, the most common boat trip for Swedes ($CO_2$ emissions are 180 kg per trip)

Our coefficient of prime interest is $b5$ –two adults without children. Given that we control for the impact on $CO_2$ emissions (from children) in the household, $b5$ represents the difference in $CO_2$ emissions from the two adults in a households, caused by adding children to the household. If $b5$ is negative, two adults emit less when childless than with children, implying that parents are browner.

Since consumption by children likely depends on the age of the child we divide children into different age groups and construct a variable that defines the number of children in each age group, on the household level. For instance, a household with two children aged 5 and 8 years, will have one child in age group 0–6 years, and one child in age group 7–12 years. Given the children within a household are divided into different age groups, the age group variables mostly consist of zeros and ones (see note in Table 2).

**Table 5. Regression results, total $CO_2$ emissions.**

| Variable | Parameter estimate | Standard error | P-value |
|----------|-------------------:|---------------:|--------:|
| Constant | 3513.80 | 323.44 | 0.000 |
| Number of children 0–6 years | 187.51 | 115.15 | 0.103 |
| Number of children 7–12 years | 422.49 | 112.80 | 0.000 |
| Number of children 13–17 years | 502.58 | 123.80 | 0.000 |
| Number of children 18–19 years | 203.51 | 218.40 | 0.351 |
| Two adults without children | -717.98 | 236.60 | 0.002 |
| One adult without children | -233.72 | 254.87 | 0.000 |
| One adult with one or more children | -1885.64 | 207.90 | 0.000 |
| Age of the oldest person in the household | 28.78 | 5.34 | 0.000 |
| Disposable income | 6.84 | 0.66 | 0.000 |
| Disposable income squared | -0.002 | 0.0003 | 0.000 |
| N | 2,680 | LM het. test = 41.66 [0.000] | |
| Adjusted R-square | 0.33 | | |

Dependent variable: Annual $CO_2$ emissions. Standard errors computed from heteroscedastic-consistent matrix. As discussed in S1 Appendix, $CO_2$ emissions from food only is based on life cycle analysis (LCA), due to scarcity of LCA $CO_2$ emissions data for other product groups. This likely means that the reported total $CO_2$ emissions in Tables 1 and 2 are on the lower end, as is our result on the difference in $CO_2$ emissions between parents and non-parents.

In the regression model we control for age of parent, given that age has been found to affect environmental preferences–older people will not live long enough to benefit from climate protection [36,37]. We also control for income since income has been shown to impact $CO_2$ emissions [15]. We include income both linearly and squared in the model, for two reasons. First, consumer demand studies based on cross sectional data note the importance of allowing the Engel curves to be non-linear [38,39]. Second, the hypothesis behind the Environmental Kuznets Curve (EKC) suggests that there is a non-linear relationship between environmental degradation and income [40,41]. Although empirical evidence on the existence of an EKC is mixed, several studies find an inverted U-shaped relationship between income and $CO_2$ emissions [41–43]. Our data enables us to contribute new and valuable information to the literature exploring the existence of the EKC. Most previous studies are based on aggregate time series or panel data for countries or regions and not on cross sectional data at the household level. Our data, in contrast, allows us to explore the relationship between income and $CO_2$ emissions (the EKC) based on substantially more detailed data.

We proceed by first using the above model to examine observed differences in total $CO_2$ emissions for adults with and without children. Using the same regression model, we thereafter explore how $CO_2$ emissions differ over consumption subgroups known to contribute heavily to $CO_2$ emissions–including food consumption (e.g., meat and milk products), transportation, and heating/electricity.

For estimation purposes we use a Least Squares estimator, i.e., an estimator that is unbiased, even if the residuals or the dependent variable are not normally distributed. We tested the dependent variable total $CO_2$ emissions from consumption for normality using a Kolmogorov-Smirnov test, and find that it is not normally distributed (test statistic = 0.039; *P*-value < 0.001).

Note that since we assume prices are the same for all households, our model might over or under estimated $CO_2$ emissions for a specific household. However, there is no systematic difference in prices across parents and non-parents. We therefore expect our econometric models presented below to generate an unbiased point estimate of the difference in $CO_2$ emissions between parents and non-parents.

## 3. Results

Three key results emerge. First, we find that two Swedish adults in a household with children emit *more* carbon than two Swedish adults in a household without children. Table 5 shows the specifics–we see 719.21 kg *less* $CO_2$ emissions for the household without children. This coefficient is substantial in magnitude and statistically significant (*P*-value = 0.001). The average couple of adults in households with children emitted 3,513.80 kg $CO_2$ annually, the childless household emitted 3,513.80–717.98 = 2,795.82. This means parents' $CO_2$ emissions are 26 percent higher than emissions by non-parent adults, which suggests that parents are *browner* than non-parents.

Table 5 further suggests that a household consisting of only one adult without children, annually emits 3,512.07–2,331.52 = 1,180.55, which is less than half of emissions of the two-adult household without children. The adult Swede with the smallest carbon footprint therefore appears to be someone who lives alone and has never had children.

Second, focusing now on children, our results suggest that each child (ages 7–17) contributes substantially to household $CO_2$ emissions, as suggested by the large coefficients with high statistical significance (*P*-value < 0.001). This confirms previous findings on the increase in $CO_2$ emissions resulting from children [2]. The impact on household $CO_2$ emissions from children age 0–6, however, is both smaller in magnitude and less statistically significant (*P*-

**Table 6. Regression results, CO₂ emissions from food.**

| Variable | Parameter estimate | Standard error | P-value |
|---|---|---|---|
| Constant | 1028.39 | 138.10 | 0.000 |
| Number of children 0–6 years | 87.73 | 46.05 | 0.057 |
| Number of children 7–12 years | 177.86 | 42.11 | 0.000 |
| Number of children 13–17 years | 365.02 | 46.66 | 0.000 |
| Number of children 18–19 years | 322.20 | 78.12 | 0.000 |
| Two adults without children | -303.97 | 90.59 | 0.001 |
| One adult without children | -924.35 | 101.78 | 0.000 |
| One adult with one or more children | -669.86 | 94.33 | 0.000 |
| Age of the oldest person in the household | 10.51 | 2.23 | 0.000 |
| Disposable income | 2.30 | 0.26 | 0.000 |
| Disposable income squared | -0.007 | 0.001 | 0.000 |
| N | 2,692 | LM het. test = 81.06 [0.000] | |
| Adjusted R-square | 0.33 | | |

Dependent variable: Annual CO₂ emissions from food consumption. Standard errors computed from heteroscedastic-consistent matrix.

value = 0.094). For most of the expenditure groups in our data, children of ages 7–17 seem to have the largest impact on CO₂ emissions, compared to younger (ages 0–6) or older children (ages 18–19). The exception is electricity and heating (see Table 10). Small children positively affect CO₂ emissions from electricity and heating, while older children either have no, or possibly a negative impact, on CO₂ emissions from this expenditure group. Both variables age of the oldest person in the household and disposable income have the expected positive impact on household CO₂ emissions, including both overall emissions and emissions from sub-groups of consumption.

Third, we find that the two major sources of consumption that explain the higher CO₂ emissions for adults with children are food and transportation. For food, Table 6 shows that

**Table 7. Regression results, CO₂ emissions from meat consumption.**

| Variable | Parameter estimate | Standard error | P-value |
|---|---|---|---|
| Constant | 295.61 | 91.64 | 0.001 |
| Number of children 0–6 years | -16.92 | 30.56 | 0. 580 |
| Number of children 7–12 years | 35.46 | 27.94 | 0.204 |
| Number of children 13–17 years | 128.37 | 30.96 | 0.000 |
| Number of children 18–19 years | 96.84 | 51.84 | 0.062 |
| Two adults without children | -121.90 | 60.11 | 0.043 |
| One adult without children | -333.47 | 67.54 | 0.000 |
| One adult with one or more children | -281.13 | 62.59 | 0.000 |
| Age of the oldest person in the household | 3.57 | 1.48 | 0.016 |
| Disposable income | 1.35 | 0.17 | 0.000 |
| Disposable income squared | -0.0004 | 0.0001 | 0.000 |
| N | 2,692 | LM het. test = 55.33 [0.000] | |
| Adjusted R-square | 0.17 | | |

Dependent variable: Annual CO₂ emissions from meat consumption. Standard errors computed from heteroscedastic-consistent matrix.

**Table 8. Regression results, CO$_2$ emissions from transportation.**

| Variable | Parameter estimate | Standard error | P-value |
|---|---|---|---|
| Constant | 1378.02 | 271.89 | 0.000 |
| Number of children 0–6 years | 88.75 | 97.08 | 0.361 |
| Number of children 7–12 years | 284.88 | 90.54 | 0.002 |
| Number of children 13–17 years | 159.58 | 104.28 | 0.126 |
| Number of children 18–19 years | 3.82 | 177.23 | 0.983 |
| Two adults without children | -396.32 | 201.47 | 0.049 |
| One adult without children | -1349.46 | 217.12 | 0.000 |
| One adult with one or more children | -1464.50 | 166.55 | 0.000 |
| Age of the oldest person in the household | 23.76 | 4.56 | 0.000 |
| Disposable income | 3.27 | 0.55 | 0.000 |
| Disposable income squared | -0.001 | 0.0002 | 0.000 |
| N | 2,692 | LM het. test = 28.10 [0.000] | |
| Adjusted R-square | 0.17 | | |

Dependent variable: Annual CO$_2$ emissions from transportation. Standard errors computed from heteroscedastic-consistent matrix.

two adults without children annually emit 303.97 kg less of CO$_2$ from food than do two adults with children (*P*-value = 0.001). Higher CO$_2$ emissions from food therefore seem to correspond to 303.97/719.21*100 = 42 percent of the total difference in CO$_2$ emissions between adults with and without children. The large share of emissions from food consumption is consistent with previous studies that demonstrate the importance of food choice to the environment [44–47].

Table 6 also shows that the two adults in a household with children annually emit 1,028.39 kg CO$_2$ from food consumption, while a corresponding household without children emits 1,028.39–303.97 = 724.42. This suggests adult CO$_2$ emissions from food is as much as 42 percent higher for parents than for adults without children. Table 7 shows that higher meat

**Table 9. Regression results, CO$_2$ emissions from gasoline.**

| Variable | Parameter estimate | Standard error | P-value |
|---|---|---|---|
| Constant | 1170.88 | 257.88 | 0.000 |
| Number of children 0–6 years | 127.99 | 97.76 | 0.190 |
| Number of children 7–12 years | 314.44 | 91.38 | 0.001 |
| Number of children 13–17 years | 177.97 | 104.07 | 0.089 |
| Number of children 18–19 years | 6.58 | 179.06 | 0.991 |
| Two adults without children | -383.57 | 203.67 | 0.060 |
| One adult without children | -1326.90 | 220.29 | 0.000 |
| One adult with one or more children | -1488.90 | 170.02 | 0.000 |
| Age of the oldest person in the household | 26.03 | 4.61 | 0.000 |
| Disposable income | 3.08 | 0.56 | 0.000 |
| Disposable income squared | -0.001 | 0.0002 | 0.000 |
| N | 2,692 | LM het. test = 26.31 [0.000] | |
| Adjusted R-square | 0.16 | | |

Dependent variable: Annual CO$_2$ emissions from gasoline. Standard errors computed from heteroscedastic-consistent matrix.

**Table 10. Regression results, CO$_2$ emissions from electricity and heating.**

| Variable | Parameter estimate | Standard error | P-value |
|---|---|---|---|
| Constant | 797.02 | 94.18 | 0.000 |
| Number of children 0–6 years | 65.89 | 31.41 | 0.036 |
| Number of children 7–12 years | -1.19 | 28.72 | 0.967 |
| Number of children 13–17 years | 20.48 | 31.82 | 0.520 |
| Number of children 18–19 years | -96.43 | 53.28 | 0.070 |
| Two adults without children | 17.71 | 61.78 | 0.774 |
| One adult without children | -42.42 | 69.41 | 0.541 |
| One adult with one or more children | 217.95 | 64.33 | 0.001 |
| Age of the oldest person in the household | 0.56 | 1.52 | 0.712 |
| Disposable income | 0.15 | 0.18 | 0.398 |
| Disposable income squared | -0.00007 | 0.00009 | 0.425 |
| N | 2,692 | LM het. test = 7.83 [0.005] | |
| Adjusted R-square | 0.008 | | |

Dependent variable: Annual CO$_2$ emissions from electricity and heating. Standard errors computed from heteroscedastic-consistent matrix.

consumption explains a large part of the higher CO$_2$ emissions from food. A two-adult household without children emits 121.90 kg less of CO$_2$ from meat consumption than do two adults with children (*P*-value = 0.043).

Compared to adults, children have weaker preferences for food that generates low CO$_2$ emissions, such as vegetables and fish [47]. Our data show that households with children have a lower budget share for fish than households without children. A potential explanation for this result might be that households converge to eating food that is agreeable to both adults and children. Previous studies find that Swedish children mostly eat breaded fish (fish sticks), [48], and that parents tend to avoid the stress associated with repeatedly offering their children foods that they are likely to reject [49].

**Table 11. Regression results, expenditures on outbound tourism (package trips).**

| Variable | Parameter estimate | Standard error | P-value |
|---|---|---|---|
| Constant | -1830.11 | 2547.79 | 0.473 |
| Number of children 0–6 years | -2788.84 | 840.11 | 0.001 |
| Number of children 7–12 years | -612.77 | 732.48 | 0.403 |
| Number of children 13–17 years | 418.05 | 846.24 | 0.621 |
| Number of children 18–19 years | 985.81 | 1518.69 | 0.516 |
| Two adults without children | 3356.52 | 1656.25 | 0.043 |
| One adult without children | 1885.98 | 1752.45 | 0.282 |
| One adult with one or more children | 1557.09 | 1510.66 | 0.303 |
| Age of the oldest person in the household | -61.52 | 37.10 | 0.097 |
| Disposable income | 50.06 | 5.68 | 0.000 |
| Disposable income squared | -0.013 | 0.003 | 0.000 |
| N | 2,692 | LM het. test = 142.91 [0.000] | |
| Adjusted R-square | 0.115 | | |

Dependent variable: Total expenditures on outbound tourism COICOP 09602 (package holidays). Standard errors computed from heteroscedastic-consistent matrix

Although regression results are not reported here, we also examined if some of higher adult $CO_2$ emissions from parents, compared to non-parent adults, could be due to higher $CO_2$ emissions from milk and cheese consumption. We found weak evidence that adult emissions from milk are higher for parents–an annual difference of 23 kg $CO_2$ (*P*-value = 0.080), while we could not reject the null that adult $CO_2$ emissions from cheese consumption are the same for parents and non-parent adults (*P*-value = 0.292).

For transportation, Table 8 shows that a two-adult household without children annually emits 397.55 kg less of $CO_2$ compared to two adults with children. This corresponds to 55 percent of the total difference in $CO_2$ emissions (= 397.55/719.21*100). A two-adult household with children annually emits 1,376.32 kg $CO_2$ from transportation, while a corresponding household without children emits 1,376.32–397.55 = 978.77. This suggests adult $CO_2$ emissions from transportation are 41 percent higher for parents than for non-parent adults. Table 9 shows that most of $CO_2$ emissions from transportation, and differences between parents and non-parents in two-adult households, result from $CO_2$ emissions from gasoline.

For electricity and heating, Table 10 shows we cannot reject the hypothesis that adult $CO_2$ emissions are the same for parents and non-parent (*P*-value = 0.774). More generally, our model fails to explain much of variations in electricity and heating, as implied by the low adjusted R-squared value (0.008). Further, this is lower than for the other models. The adjusted R-squared values suggest that the regression models for total $CO_2$ emissions and emissions from food have the highest explanatory power, with adjusted R-squared values of 0.33. The models for emissions from meat, transportation and gasoline have adjusted R-squared values of around 0.17, suggesting there is more heterogeneity in emissions from these goods.

Our results remain unchanged if we exclude clothes and footwear, suggesting this expenditure group both has a minor impact on overall $CO_2$ emissions, and that parents' $CO_2$ emissions from clothes and footwear in two-adult households do not differ from those of non-parents.

In all regression models we also find a statistically significant inverted U-shaped relation between income and $CO_2$ emissions, which supports the EKC hypothesis. The only exception is the model for $CO_2$ emissions from electricity and heating, where the point estimates for the income variables are insignificant. We also estimated all regression models including higher order polynomials of disposable income, which, however, turned out to be statistically insignificant at any commonly used significance level. Our unique micro level data set therefore confirms the previous finding of an Environmental Kuznets Curve for Swedish total $CO_2$ emissions [37, 50].

We considered the robustness of our main results by addressing three aspects of sensitivity–age of household, significance uncertainty as measured by four sources of variability, and expenditures on package trips. For age, we exclude retired subjects to reduce 'contamination' of our data for households without children from childless households with adult children. In the robustness checks we lower the age of the oldest adult in the household to 65, 55 and 45 year. Although our total sample size drops when reducing the age, we see no change in our basic result that parents have a larger carbon footprint. S2 Appendix provides the details.

For uncertainty, we examine four potential sources of variability: (a) $CO_2$ emissions from boat trips (emissions from a cruise Stockholm-Helsinki), (b) energy usage from district heating (15 percent of rental costs), (c) $CO_2$ emissions from gasoline (2.24 kg/liter gasoline), and (d) the price of public transportation (calculated as long-distance trips, suggesting the price may be on the lower end). Overall, we find no substantial effect on our main results in any of the four sources of uncertainty–again our results remain robust (see S2 Appendix for specifics).

For expenditures on package trips, Table 11 shows that a two-adult household without children spends SEK 3,356.52 more on international package trips annually compared to two adults in a household with children (*P*-value = 0.043). The expenditure data entails no

information on the content of international package trips. However, travel surveys indicate that about one third of household expenditures on domestic trips are spent on transportation [51]. If the same proportion is spent on transportation for international package trips, this suggests that a two-adult household without children emits 207.6 kg more of $CO_2$ compared to two-adults in a household with children, if making the assumption that all travel related expenditures are spent on gasoline. This implies our main results hold even if parents may emit less from international trips, compared to non-parents. S2 Appendix provides more information about the data and assumptions for Swedes' international trips and $CO_2$ emissions.

## 4. Concluding remarks

Becoming a parent can transform a person–he or she thinks more about the future and worries about future risks imposed on their children and progeny. The open question that we explore is whether this transformation might imply a person will be greener. Do parents have a smaller carbon footprint? Using a unique data set that allows us to compare $CO_2$ emissions from Swedish two-adult households with and without children, we find that two adults in households with children increase $CO_2$ emissions by more than 25 percent relative to two adults in households without children. Parents' $CO_2$ emissions are higher due to increased transportation and changed food consumption. Although having children might be transformational, our results suggest parents' "new" preferences do not cause them to be greener than non-parent adults.

We speculate that our Swedish household results may represent a lower bound on the parental carbon footprint relative to most other Western countries. Most Swedes believe climate change is a reality such that becoming more forward looking should entail caring more about climate change, and the Swedish government subsidizes childrearing to help reduce stress and increased time constraints from parenthood.

While the benefits of our data are their level of detail and the broad range of consumption types covered, a drawback is that the cross-sectional nature of the data disables analysis of causality between parenthood and the observed higher carbon emissions. That being said, self-selection into parenthood based on preference greenness is unlikely: self-selection into parenthood itself is rare [22], and self-selection based on green preferences even more so, to the extent that previous studies fail to detect any effect of the decision to become a parent from factors either directly or indirectly affecting $CO_2$ emissions [23]. It is therefore not unreasonable to imagine that our results imply a causal link–parenthood *makes* people browner. The larger footprint from parenthood arises mainly from higher transportation emissions and browner food.

What causes parents to emit more of $CO_2$? The higher $CO_2$ emissions from parenthood could be the net outcome of two previously documented counter-veiling effects: parents care more for the future [4,5], while being more pressed to satisfy immediate needs under tighter time constraints [24]. In this study, we cannot directly observe these underlying consumption motivations. While our study suggests the difference in $CO_2$ emissions between parents and non-parents is substantial, and therefore warrants policy makers' attention, we encourage future research to examine the strength of the underlying factors that might cause the consumption differences. Such knowledge might be important in identifying policies that effectively reduce the carbon footprint from parents, e.g., policies that alleviate their time crunch–increased access to child care, increased work hour flexibility for parents, and so on.

Further, the differences in food consumption between parents and non-parents might be helped by recent innovations in developing meat substitutes that "taste like meat," and hence might appeal to everyone in the family [52]. Such meat substitutes are still relatively expensive

—government subsidies of meat substitutes could help speed up the transition from meat to vegetarian proteins, especially for families with children, given they might be particularly constrained by satisfying heterogonous food preferences within the family.

Finally, our results might also imply that there could be distributional effects from the Swedish carbon tax across families with and families without children, when controlling for the number and age of people in the family. A follow-up study might determine the incidence of carbon taxes across households with and without children.

## Supporting information

**S1 Appendix. Data.**
(DOCX)

**S2 Appendix. Robustness checks.**
(DOCX)

## Author Contributions

**Conceptualization:** Jason F. Shogren, Linda Thunström.

**Data curation:** Jonas Nordström.

**Formal analysis:** Jonas Nordström, Jason F. Shogren, Linda Thunström.

**Methodology:** Jonas Nordström, Linda Thunström.

**Writing – original draft:** Jason F. Shogren, Linda Thunström.

**Writing – review & editing:** Jonas Nordström, Jason F. Shogren, Linda Thunström.

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
