## [Decision Letter · Decision Letter 0]

21 Jan 2020

PONE-D-19-31142

Do parents counter-balance the carbon emissions of their Children?

PLOS ONE

Dear Dr Nordström,

Thank you for submitting your manuscript to PLOS ONE. After careful consideration, we feel that it has merit but does not fully meet PLOS ONE’s publication criteria as it currently stands. Therefore, we invite you to submit a revised version of the manuscript that addresses the points raised during the review process.

The authors should engage in a major review of the manuscript and econometric model that addresses all observations raised by Reviewer 2. We would appreciate receiving your revised manuscript by Mar 06 2020 11:59PM. To enhance the reproducibility of your results, we recommend that if applicable you deposit your laboratory protocols in protocols.io, where a protocol can be assigned its own identifier (DOI) such that it can be cited independently in the future. For instructions see: http://journals.plos.org/plosone/s/submission-guidelines#loc-laboratory-protocols

We look forward to receiving your revised manuscript.

Kind regards,

Francisco X Aguilar

Academic Editor

PLOS ONE

Journal Requirements:

2. In order to enable reproducibility and replicability, please ensure you have described in detail how other researchers could obtain the same data used in your study from Statistics Sweden.

Additional Editor Comments:

Two reviewers have offered their assessment to this submission. While one suggested its acceptance as it is, the second reviewer offered a detailed and constructive criticism of this empirical application. Among the reviewer's observations I highlight the motivation for this study, additional comments regarding the limitation of the data and econometric model, and the consideration of additional explanatory variables within their OLS regression.

All of Reviewer 2's comments are constructive, hence, addressing them will result in a much improved version. For instance, it seems that within the current model the authors could also easily control for rural/urban effects. As pointed by Reviewer 2 as well it seems the current model does not quiet match the profile of a behavioral one. For instance, one would expect attitudes, social norms, among other constructs to help explain the actual behavior in this case.  The authors should reconsider labeling their model this way.  Some additional details on the model estimation should be provided such as the test-statistic used to justify the use of robust standard errors. The authors should also offer a test for normality in the distribution of their dependent variable. Why the reporting of 'adjusted' R-squared if the model seemed to be estimated using OLS? A word on the limitations of the econometric model, inclusive of relatively low R-squared values, is warranted.

I will also ask the authors to possibly refrain from suggesting that the effects tested are the result of "becoming a parent".  Such choice of words infers a degree of change in emissions by the same individual (or household) before/after children in a more traditional differences-in-differences approach or in some kind of panel data setting - which is certainly not the case in the current dataset.

Reviewers' comments:

Reviewer's Responses to Questions

**Comments to the Author**

1. Is the manuscript technically sound, and do the data support the conclusions?

Reviewer #1: Yes

Reviewer #2: Partly

2. Has the statistical analysis been performed appropriately and rigorously? 

Reviewer #1: Yes

Reviewer #2: Yes

3. Have the authors made all data underlying the findings in their manuscript fully available?

Reviewer #1: No

Reviewer #2: No

4. Is the manuscript presented in an intelligible fashion and written in standard English?

Reviewer #1: Yes

Reviewer #2: Yes

5. Review Comments to the Author

Reviewer #1: I recognize that the authors are using publicly accessible secondary data that they can't include as part of their paper, and in my view that's not really a major problem as long as the data _remain_ publicly accessible, but I leave it to the editors to judge how best to deal with this if the paper is accepted for publication.

Reviewer #2: Review comments to the authors are included in a separate attached file.

Concerning data my understanding is that the underlying data is not available for others without special permission from Statistics Sweden. This is also explained in the manuscript pdf-file.

6. PLOS authors have the option to publish the peer review history of their article (what does this mean?). If published, this will include your full peer review and any attached files.

Reviewer #1: No

Reviewer #2: No

---

## [Author Response · Author response to Decision Letter 0]

12 Mar 2020

Comments to the reviewers and editor are attached in the word document named "Response to Reviewers"

---

## [Editor Report · Decision Letter 1]

17 Mar 2020

Do parents counter-balance the carbon emissions of their Children?

PONE-D-19-31142R1

Dear Dr. Nordström,

We are pleased to inform you that your manuscript has been judged scientifically suitable for publication and will be formally accepted for publication once it complies with all outstanding technical requirements.

With kind regards,

Francisco X Aguilar

Academic Editor

PLOS ONE

---

## [Editor Report · Acceptance letter]

20 Mar 2020

PONE-D-19-31142R1 

Do parents counter-balance the carbon emissions of their Children? 

Dear Dr. Nordström:

I am pleased to inform you that your manuscript has been deemed suitable for publication in PLOS ONE. Congratulations! Your manuscript is now with our production department. 

With kind regards,

on behalf of

Dr. Francisco X Aguilar 

Academic Editor

PLOS ONE